# Oxygen Vacancies in Bismuth Tantalum Oxide to Anchor Polysulfide and Accelerate the Sulfur Evolution Reaction in Lithium–Sulfur Batteries

**DOI:** 10.3390/nano12203551

**Published:** 2022-10-11

**Authors:** Chong Wang, Jian-Hao Lu, An-Bang Wang, Hao Zhang, Wei-Kun Wang, Zhao-Qing Jin, Li-Zhen Fan

**Affiliations:** 1Beijing Advanced Innovation Center for Materials Genome Engineering, Institute of Advanced Materials and Technology, University of Science and Technology Beijing, Beijing 100083, China; 2Military Power Sources Research and Development Center, Research Institute of Chemical Defense, Beijing 100191, China

**Keywords:** lithium–sulfur battery, oxygen vacancies, electrochemical performance, high areal mass loading

## Abstract

The shuttling effect of soluble lithium polysulfides (LiPSs) and the sluggish conversion kinetics of polysulfides into insoluble Li_2_S_2_/Li_2_S severely hinders the practical application of Li-S batteries. Advanced catalysts can capture and accelerate the liquid–solid conversion of polysulfides. Herein, we try to make use of bismuth tantalum oxide with oxygen vacancies as an electrocatalyst to catalyze the conversion of LiPSs by reducing the sulfur reduction reaction (SRR) nucleation energy barrier. Oxygen vacancies in Bi_4_TaO_7_ nanoparticles alter the electron band structure to improve instinct electronic conductivity and catalytic activity. In addition, the defective surface could provide unsaturated bonds around the vacancies to enhance the chemisorption capability with LiPSs. Hence, a multidimensional carbon (super P/CNT/Graphene) standing sulfur cathode is prepared by coating oxygen vacancies Bi_4_TaO_7−x_ nanoparticles, in which the multidimensional carbon (MC) with micropores structure can host sulfur and provide a fast electron/ion pathway, while the outer-coated oxygen vacancies with Bi_4_TaO_7−x_ with improved electronic conductivity and strong affinities for polysulfides can work as an adsorptive and conductive protective layer to achieve the physical restriction and chemical immobilization of lithium polysulfides as well as speed up their catalytic conversion. Benefiting from the synergistic effects of different components, the S/C@Bi_3_TaO_7−x_ coin cell cathode shows superior cycling and rate performance. Even under a high level of sulfur loading of 9.6 mg cm^−2^, a relatively high initial areal capacity of 10.20 mAh cm^−2^ and a specific energy density of 300 Wh kg^−1^ are achieved with a low electrolyte/sulfur ratio of 3.3 µL mg^−1^. Combined with experimental results and theoretical calculations, the mechanism by which the Bi_4_TaO_7_ with oxygen vacancies promotes the kinetics of polysulfide conversion reactions has been revealed. The design of the multiple confined cathode structure provides physical and chemical adsorption, fast charge transfer, and catalytic conversion for polysulfides.

## 1. Introduction

The growing energy needs of modern society have boosted the development of clean and sustainable energy sources. Lithium-Sulfur (Li-S) batteries are considered a potential candidate for next-generation energy-storage devices because of their excellent merits such as high energy density, low price, and an abundance of sulfur resources [1,2]. However, the implementation of Li-S batteries is still hindered by several issues, including the inherent low conductivity of sulfur and Li_2_S and the shuttling behavior caused by the dissolution of lithium polysulfide (LiPSs) in the electrolyte; large volumetric change, resulting in poor utilization of active sulfur; low coulombic efficiency; and poor recycling ability [3,4].

In the past few years, different research has been carried out to solve these problems. Carbonaceous materials can enhance the conductivity of a cathode, while the Van der Waals force between nonpolar carbon and polar LiPSs is unable to effectively inhibit the undesirable LiPSs’ shuttling due to weak physical confinement [5,6,7]. Apart from carbon materials, catalysis processes such as transition metal oxide, sulfide, and phosphide have also been explored to catalyze the electrochemical conversion of lithium polysulfides [8,9,10,11,12,13,14,15]. However, bulk atoms in a homogeneous and continuous environment are usually less active due to the saturated bonding state. These electrocatalyst activities can be further improved through defect engineering by altering their electronic structures [16,17]. Their specific electronic states offer them enhanced LiPSs with adsorptive and catalytic features. However, a detailed explanation of the mechanism by which anion vacancies enhance the Li-S battery performance is still required [18,19].

The semiconductor material, bismuth tantalum oxide (Bi_3_TaO_7_), has the possibility to be applied in Li-S batteries, because the vacant orbitals of Bi could easily provide electroactive sites and accept external electrons from polysulfides. However, its relatively low catalytic activity and conductivity limits its application as a catalytic material of the sulfur electrode for Li-S batteries [20]. Herein, Bi_3_TaO_7_ with oxygen vacancies is used as an example to establish the relationship between oxygen vacancy and adsorptive–catalytic properties. Based on the above analysis, we developed a multidimensional carbon standing sulfur cathode by cooperating with Bi_3_TaO_7−x_ with oxygen vacancies, in which the multidimensional carbon offers enough space to host sulfur and a continuous conductive structure. Additionally, Bi_3_TaO_7_ with oxygen vacancies richly distributed on the surface of S/MC can inhibit the polysulfide shuttling effect and provide catalytically active sites. As expected, the S/C@Bi_3_TaO_7−x_ coin cell presents a low-capacity fading rate of 0.047% per cycle over a long-term operation of 500 cycles. Furthermore, the pouch-type cell with the S/C@Bi_3_TaO_7−x_ cathode attains a high energy density of about 300 W h kg^−1^. DFT calculations also reveal the mechanism by which oxygen vacancies could increase the binding energy of polysulfides and enhance the catalytic activity toward the sulfur reduction reaction. This work provides deep insight into understanding the adsorptive and catalytic properties of defective catalytic materials in Li-S batteries.

## 2. Experimental

### 2.1. Synthesis of Bi_3_TaO_7−x_

Bi_3_TaO_7_ was prepared using a simple hydrothermal method. First, 0.001 mol of TaCl5(Aladdin) was dissolved in 60 mL of ethanol, and then, 0.003 mol of Bi (NO_3_)_3_·5H_2_O(Aladdin) was added to the above ethanol solution and stirred [21]. The pH of the suspension was adjusted to 10 using a potassium hydroxide (Aladdin) solution. Then, the suspension was transferred to a hydrothermal reactor and dried at 230 °C overnight. Subsequently, the Bi_3_TaO_7_ powers were annealed at 650 °C in Ar/H_2_ gas flow to form oxygen vacancies.

### 2.2. Preparation of S/C@Bi_3_TaO_7−x_ Cathodes

Firstly, a certain amount of super P, CNT, and Graphene (with the weight ratio of super P:CNT:G = 2:2:1) was mixed uniformly by using a hand mill to form the multidimensional carbon (MC). Then, the MC composites were dispersed in deionized (DI) water under stirring for 3h to form a homogeneous slurry. Elemental sulfur was synthesized based on the reaction between Na_2_S_2_O_3_ and HCOOH. The elemental sulfur solution was slowly added into the aqueous solution of super P, CNT, and Graphene under continuous stirring at 60 °C for 10 h. Then, the sediment of the S/MC was filtered, washed repeatedly with deionized water to remove those soluble impurities, and dried under a vacuum of 60 °C for 24 h. The sulfur content of the S/MC composite was about 85 wt%.

The S/C@Bi_3_TaO_7−x_ composite with 10% Bi_3_TaO_7−x_ was prepared via the electrostatic self-assembly process [22], and the as-prepared S/MC composite material was dispersed in 1 L of 0.5wt% cetyltrimethylammonium bromide (CTAB) aqueous solution and stirred for 24 h. Then, 300 mL of the homogeneous aqueous dispersion of Bi_3_TaO_7−x_ was dipped into the S/C@CTAB solution and stirred for 3 h. To obtain the S/C@Bi_3_TaO_7−x_ composite, the precipitate was filtered, washed with deionized water, and dried at 60 °C.

### 2.3. Material Characterization Techniques

The sulfur content of S/C@Bi_3_TaO_7−x_ and S/MC was tested using thermogravimetric analysis (Netzsch Inc., Selb, Germany) Scanning electron microscopy (Carl Zeiss Inc., Oberkochen, Germany) was employed to examine the morphologies of the samples. Transmission electron microscopy (TEM) and element mapping measurements were performed on an FEI Talos 200s microscope(Thermo Fisher Scientific Inc., Waltham, MA, USA). The chemical analysis of the as-prepared samples was conducted using X-ray photoelectron spectroscopy (XPS) and X-ray diffraction (Rigaku Inc., Tokyo, Japan). A Perkin Elmer Lambda 950 UV–vis spectrophotometer(Perkin Elmer Inc., Waltham, MA, USA) was used to measure UV–vis absorption spectra. Raman spectrometry was recorded on a LabRAM HR Evolution (HORIBA Inc., Kyoto, Japan). Oxygen vacancy was determined using electronic paramagnetic resonance (Bruker Inc., Rheinstetten, Germany). The surface areas were obtained using Brunauer–Emmett–Teller analysis of the adsorption isotherm.

### 2.4. Visualized Adsorption Experiment

The Li_2_S_6_ solution was prepared by mixing lithium sulfide (Li_2_S) and elemental sulfur with a mass ratio of 1:5 into the mixed solvent (1,3-dioxolane and 1,2-dimethoxyethane, *v*/*v* = 1:1) and stirred for 12 h. Then, 30 mg of MC (multidimensional carbon), Bi_3_TaO_7,_ and Bi_3_TaO_7−x_ were put into 30 mL of Li_2_S_6_ solution and rested for 12 h. The supernatant liquid was studied via UV–vis spectrophotometry.

### 2.5. Symmetrical Cell Measurement and the Li_2_S Nucleation Test

The symmetric cells were assembled with aluminum-foil-loaded Bi_3_TaO_7_, Bi_3_TaO_7−x_, and MC as both the cathode and anode. Cyclic voltammetry (CV) and electrochemical impedance spectroscopy (EIS) measurements were conducted on a VersaSTAT3 electrochemical workstation. CV was performed at a scan rate of 50 mV/s with a voltage window between −1V and 1V. EIS was performed at open circuit potential with a frequency range from 10 mHz to 100 kHz with an amplitude of 10 mV.

Li_2_S_8_ solution was obtained by mixing sulfur and Li_2_S in a molar ratio of 1:7 in DOL/DME (1:1, *v*/*v*) solution and vigorously stirred overnight. The preparation process of the electrode can be referred to in the description of symmetrical batteries. The lithium foil was used as the anode, and Celgard 2500 was used as the separator. The 25 μL Li_2_S_8_ solution, together with 25 μL of 1 M LiTFSI + 2wt%LiNO_3_ dissolved in DOL/DME (1:1 *v*/*v*), was added as an electrolyte. To carry out Li_2_S nucleation tests on different substrates, the assembled cells were firstly galvanostatically discharged (0.134 mA) to 2.06 V, followed by being potentiostatically kept at 2.05 V for Li_2_S to nucleate until the current density decreased to 10^−5^ A.

### 2.6. Theoretical Calculations

Density functional theory (DFT) calculations were performed by the Dmol3 module of Materials Studio, using the generalized gradient approximation method with the Perdew–Burke–Ernzerhof function. The convergence criteria for residual force and energy were set to 0.03 eV/Å and 10^−5^ eV. For the adsorption conformation simulations, DFT-D of dispersion correction was adopted to describe the van der Waals (vdW) interactions. The Brillouin zone of the supercell was sampled by a parameter of 2 × 2 × 1. The adsorption energy was calculated as E_B_ = E_(Surf − Li2Sx)_ − E_Surf_ − E_Li2Sx_. The Gibbs free energy can be expressed through the following equation: ∆G = ∆E*_DFT_* + ∆Z*_PE_* − T∆S [23].

### 2.7. Cell Assembly/Electrochemical Measurement

The Li-S battery cathodes were prepared by coating the black slurry of the composite on an aluminum foil collector. First, LA133 (10 wt%) as a binder was dispersed into deionized water and isopropanol solution. Then, 80 wt% of the active material, 5 wt% super P, and 5 wt% carbon nanotube as the conductive host were added to the above solution for stirring. Next, the cathode slurry was blade cast onto the current collector Al foils by an automatic coating machine and then dried. The electrode was punched into 12 mm, and the mass loading of sulfur was controlled to be 1.5 mg cm^−2^. The specific capacities were calculated according to the loading mass of sulfur in the cathodes.

The standard CR 2025 coin cell was assembled with the obtained cathode electrode, with Celgard 2400 polypropylene membrane as the separator, lithium metal as the anode, and 1M LiTFSI + 2 wt%LiNO_3_ dissolved in DOl/DME (1:1 *v*/*v*) as the electrolyte. The electrolyte/sulfur (E/S) ratio was about 20 µL mg^−1^. The charge–discharge cycling of coin cells was tested between 1.8 V and 2.6 V using a LAND CT2001A multi-channel battery testing system. CV measurements were performed on the VersaSTAT3 electrochemical workstation.

## 3. Results and Discussion

The S/C@Bi_3_TaO_7−x_ was fabricated using a simple solution-phase electrostatic self-assembly method, illustrated in Figure 1. Due to the effect of oxygen vacancies on Zeta potential, the Zeta potentials of Bi_3_TaO_7_ and Bi_3_TaO_7−x_ at pH 7 changed from −11.5 mV to −24.1 mV (Appendix A). As a result of the positive zeta potential for sulfur composite materials (SCMs) in deionized water, Bi_3_TaO_7−x_ could evenly distribute on the surface of SCM via electrostatic attraction [24].

The morphology and microstructure of the Bi_3_TaO_7−x_ were conducted employing SEM measurements, the synthesized sample displayed egg-like configurations with an average diameter of about 500 nm in Figure 2a. The hexagonal crystalline feature of Bi_3_TaO_7−x_ was investigated through a high-resolution TEM image and the inverse fast Fourier transform (FFT) pattern. Regular crystalline fringes can be observed in Figure 2b; the distinct lattice fringes of 0.28 nm and 0.32 nm are consistent with the (211) and (111) planes of Bi_3_TaO_7_ [19]. On the contrary, the introduction of oxygen vacancies makes some defects and amorphous Bi_3_TaO_7−x_. We can see that in the inverse fast Fourier transform, the image of Bi_3_TaO_7−x_ exhibits a disordered lattice [25], which demonstrates that the arrangement of the crystal plane was severely damaged by oxygen vacancies in Figure 2c. The EDS elemental mapping of Bi_3_TaO_7−x_ reveals that Bi, Ta, and O elements had homogeneous distribution across the analyzed region in Figure 2d–f. The pore structures of Bi_3_TaO_7−x_ were studied via nitrogen adsorption–desorption analysis; it showed a specific surface area of 497.8 m^2^ g^−1^ and the coexistence of micropores and mesopores. A porous coating layer is beneficial for the infiltration of electrolytes to the active material and confinement of soluble polysulfides (Appendix A). SEM images of S/C@Bi_3_TaO_7−x_ and the EDS elemental mapping of S/C@Bi_3_TaO_7−x_ and S/C are shown in Appendix A; they can confirm that Bi_3_TaO_7−x_ particles are successfully covered on the surface of S/C. A Bi_3_TaO_7−x_ coating could provide a ‘‘physical and chemical” dual restriction function to anchor the LiPSs and catalyze their conversion. Thermogravimetric analysis (TGA) revealed that the S content of S/C@ Bi_3_TaO_7−x_ was about 75% (Appendix A).

X-ray diffraction (XRD) patterns show that all two samples have a similar hexagonal structure (JCPDS No.44-0202) [21], and the defect regulation makes no difference in the basic crystalline phase. However, a very slight shift to a high angle can be seen when comparing the diffraction peaks (200), (220), and (222) in Figure 3a, indicating the lattice parameters’ enlargement, mainly due to the existence of oxygen vacancies. At the same time, the electron paramagnetic resonance (EPR) measurement was performed to verify the existence of oxygen vacancies. Bi_3_TaO_7−x_ shows a considerable signal with a guess value of 2.001 in Figure 3b [25], suggesting the existence of unpaired electrons formed by the absence of partial oxygen atoms. In addition, Raman peaks in the frequency range of 150–300 cm^−1^ are due to the bending mode of axial Ta–O–Ta bonds. The peak is weakened or disappears due to the H_2_ reduction process, indicating the partial oxygen atoms in the crystal are removed to form oxygen vacancies in Figure 3c. As shown in Figure 3d, the XPS analysis was performed to probe the existence of an oxygen vacancy in Bi_3_TaO_7−x_; the O^1S^ spectrum of Bi_3_TaO_7−x_ could be deconvoluted into three characteristic peaks located at 529.0 eV (peak I), 530.5 eV (peak II), and 532.1 eV (peak III). Peak I can correspond to lattice oxygen, peak II is attributed to the abundant defect sites with lower oxygen coordination (oxygen vacancy), while peak III is related to surface-absorbed water [26]. The shift in the peak Ta 4f in Bi_3_TaO_7−x_ demonstrates a change in the surrounding coordination environment (Appendix A). Notably, the XPS results further suggest that the oxygen vacancies were successfully synthesized. The adsorption visualization experiment was used to study the adsorptive property, equal amounts of Bi_3_TaO_7−x_ and Bi_3_TaO_7_ were immersed in the Li_2_S_6_ solution, and the Bi_3_TaO_7−x_ containing LiPS solution appeared transparent after aging for 12 h in comparison to the initial state of the Li_2_S_6_ solutions, signifying its higher discoloration capability compared with Bi_3_TaO_7_. Consistently, the absorbance was also analyzed via UV−vis adsorption spectra; the Bi_3_TaO_7−x_ solution witnessed a drastic decline in absorbance intensity, as shown in Figure 3e. The enhanced LiPS adsorption capability could be attributed to the existence of unsaturated bonds around vacancies and the polar surface rearrangement [27].

Figure 4a shows the valence band XPS spectra for Bi_3_TaO_7_, whose maximum energy is located at 1.25 eV, while the VB maximum of Bi_3_TaO_7−x_ moves toward 0.76 eV. The formation of new electronic states induced by oxygen vacancies could explain the blue shift. The Kubelka–Munk plot shows that the band-gap values of Bi_3_TaO_7−x_ and Bi_3_TaO_7_ are 3.83 eV and 4.19 eV, as shown in Figure 4b. The electron band diagram is illustrated in Figure 4c; Bi_3_TaO_7−x_ presents a lower CBM than that of Bi_3_TaO_7_, which clarifies that oxygen vacancies could narrow the band gap of Bi_3_TaO_7_ to strengthen the electrical conductivity and make electron transfer faster [28]. To further explore the relationship between the oxygen vacancies and electrical conductivity, the four-point probe method was used to determine the electrical conductivities of Bi_3_TaO_7−x_ and Bi_3_TaO_7_. Bi_3_TaO_7−x_ possesses a much higher conductivity of 3.4 × 10^−^^2^ S m^−^^1^ than pristine Bi_3_TaO_7_ with 5.1 × 10^−7^ S m^−^^1^, as shown in Figure 4d. It is clear that oxygen vacancies play a crucial role in improving the intrinsic electrical conductivity of Bi_3_TaO_7_ [28].

The electrocatalytic performance of Bi_3_TaO_7−x_, Bi_3_TaO_7_, and the MC reference was investigated using a series of electrochemical tests. To analyze the electrocatalytic activity of Bi_3_TaO_7−x_ in the liquid–liquid process, CV curves of Li_2_S_6_ symmetrical cells are displayed in Figure 5a [29]. The Bi_3_TaO_7−x_ exhibits the highest redox current response of 20 mA among these samples. In addition, the Bi_3_TaO_7−x_ electrode delivers the highest exchange current density in both reduction and oxidation processes and the smallest Tafel slope at the same time, as shown in Figure 5b [30]. Compared with those of Bi_3_TaO_7_ and MC, the electrochemical impedance spectroscopy (EIS) measurement results show that Bi_3_TaO_7−x_ possesses the lowest charge-transfer resistance (22 Ω) when compared with those of Bi_3_TaO_7_ (46 Ω) and MC (84 Ω), indicating its fast charge transfer and enhanced polysulfide redox kinetics, as shown in Figure 5c [31]. Finally, the kinetic analysis of liquid–solid conversion was studied using Li_2_S deposition experiments. A good kinetic of liquid–solid conversion is vital to guarantee capacity contribution. The chronoamperometry curves indicate that Bi_3_TaO_7−x_ has the shortest incubation time (470 s) and the highest Li_2_S nucleation capacity (210.9 mAh g^−1^) compared to that of Bi_3_TaO_7_ and MC, as shown in Figure 5d–f. This can be attributed to the fact that Bi_3_TaO_7−x_ with oxygen vacancies can reduce the energy barrier for Li_2_S nucleation and accelerate the conversion redox kinetics, indicating their superior electrocatalytic effect on boosting the conversion from liquid LiPSs to solid Li_2_S [32].

The different electrodes were prepared and measured to explore the application potential for Li-S batteries. Figure 6a shows the CV curves of three electrodes in a voltage range of 1.7 to 2.8 V at a scan rate of 0.1 mV s^−1^ [33]. It was found that the S/C@Bi_3_TaO_7−x_ cathode displays two typical cathodic peaks at 2.03 V (peak i) and 2.35 V (peak ii), which correspond to the reduction of sulfur into high-order soluble LiPSs and the subsequent conversion of the LiPSs to insoluble Li_2_S_2_/Li_2_S. The anodic peaks at 2.38 V (peak iii) are associated with reverse oxidation conversion from Li_2_S to LiPSs and finally to sulfur. Compared with S/C@Bi_3_TaO_7_ and MC, S/C@Bi_3_TaO_7−x_ exhibits a higher reduction peak potential and lower oxidation peak potential. Moreover, the S/C@Bi_3_TaO_7−x_ cathode also delivers the highest discharge capacity of 1250 mAh g^−1^ and the smallest potential gap between charge and discharge plateaus at 0.1C, as shown in Figure 6b [34], indicating the improved conversion reaction kinetics. As shown in Figure 6c, S/C@Bi_3_TaO_7−x_ unveils the best rate capability with a decent discharge capacity of 780.2 mAh g^−^^1^ at the current rate of 3 C in contrast to S/C@Bi_3_TaO_7_ 350.6 mAh g^−1^ and S/C 280.5 mAh g^−^^1^, and the reversible capacity of 977 mAh g^−1^ when the current returns to 0.2 C. Appendix A presents the galvanostatic charge/discharge profiles of S/C@Bi_3_TaO_7−x_ at different current rates, all discharge curves of S/C@Bi_3_TaO_7−x_ display two well-defined plateaus with small polarization potentials, even at the highest current density tested at the current rate of 3 C. The cycling performances of S/C@Bi_3_TaO_7−x,_ S/C@Bi_3_TaO_7_, and S/C are checked at a constant current density of 0.5 C to further evaluate the long-term stability in Figure 6d; the initial specific capacity of S/C@Bi_3_TaO_7−x_ is 985 mAh g^−1^ with a capacity decay rate of 0.047% per cycle within 500 cycles. The excellent cyclability and rate capability of S/C@Bi_3_TaO_7−x_ benefit from the advanced host material design, the physical confinements to suppress the LiPSs’ shuttle effect, the highly electron/ion-conductive structure of multidimensional carbon, and highly effective sulfur immobilization and catalyzation via Bi_3_TaO_7−x_ rich in oxygen vacancies.

To explore the practical application potential of S/C@Bi_3_TaO_7−x_, the pouch cells included six 5.6 × 7.2 cm cathode slices with high sulfur loading, a thin Li anode, and lean electrolyte operation, which are shown in Figure 6e [35]. The S/C@Bi_3_TaO_7−x_ electrode with high sulfur loading delivers a high areal capacity of 10.20 mAh cm^−2^ under the high sulfur loading condition of 9.6 mg cm^−2^ and a low E/S ratio of 3.3 µL mg^−^^1^ with a high practical specific energy of 300 Wh kg^−1^ (Appendix A). Meanwhile, an LED light can be easily lit up using an S/C@Bi_3_TaO_7−x_ -assembled pouch cell, suggesting a practical application for powering electronic devices.

We summarize many of the previously reported results from other Li-S pouch cells, our Li-S pouch cell represents a significant advance among other recently published electrodes (Appendix A). According to these results, the enhanced electrocatalytic activity for sulfur species is due to the unique structure of the S/C@Bi_3_TaO_7−x_ electrodes. The multidimensional carbon provides physical confinement for the substance sulfur and anchors polysulfides to prevent lateral diffusion. Moreover, Bi_3_TaO_7−x_ with oxygen vacancies could accelerate surface electron exchange for a fast redox reaction and increase the binding energy of polysulfides.

The effect of oxygen vacancies with Bi_3_TaO_7_ on enhancing the adsorptive and catalytic capability of Li-S batteries was investigated via density functional theory (DFT). Vacancy Engineering is an effective method to alter a surface electronic environment through the local redistribution of electrons. Furthermore, the introduction of surface defects could offer unsaturated bonds around vacancies to provide catalytically active sites [36]. Analyzing the charge density of Bi_3_TaO_7_ and Bi_3_TaO_7−x_ in Figure 7a, the existence of the oxygen vacancies is shown, which are near Bi, Ta atoms. As shown in Figure 7b, there is a narrow gap near the Fermi lever after the formation of oxygen vacancies, indicating that Bi_3_TaO_7_ is a semiconductor with poor electrical conductivity.

In Figure 7c, some electrons are distributed near the Fermi level; this is evidence that oxygen vacancies could enhance electron mobility and catalytic activity by forming new electronic states located in the band gap [37]. Figure 7d shows the calculated binding energies between two electrocatalysts and LiPSs; the Bi_3_TaO_7−x_ delivers a stronger adsorption capacity towards LiPSs than that of Bi_3_TaO_7_, which is consistent with visualized adsorption testing. The strong polysulfide adsorption capacity makes it easy to break the S–S and Li–S bonds and improves redox reaction kinetics.

During the sulfur reduction reaction (SRR), the Gibbs free energy of the intermediates is highly related to the activity of the electrocatalyst on LiPSs’ conversion [38]. Figure 7e shows the Gibbs free energy profile from S_8_ to Li_2_S on the Bi_3_TaO_7−x_ and Bi_3_TaO_7_ surfaces; the calculated results show that the electrochemical reduction conversion of S is spontaneous exothermic conversion, indicating that the presence of Bi_3_TaO_7−x_ effectively decreases the energy barrier of the solid–solid conversion reaction.

These results demonstrate that the introduction of oxygen vacancy can result in narrowing the bandgap to enhance the intrinsic conductivity and kinetics of LiPSs conversion reactions when compared with those of the corresponding parent materials. Furthermore, the existence of oxygen vacancies also improves the adsorption ability of Bi_3_TaO_7−x_ for LiPSs. In contrast to Bi_3_TaO_7_, Bi_3_TaO_7−x_ with oxygen vacancies exhibits excellent adsorptive–catalytic properties. DFT calculations give us a deep understanding of the electronic structure and catalytic mechanism of Bi_3_TaO_7−x_ in Li-S electrochemistry.

## 4. Conclusions

In summary, a multi-component structure with multidimensional carbon and oxygen vacancies Bi_3_TaO_7−x_ was constructed, which affords the synergistic functions of physical confinement, chemical anchoring, and excellent electrocatalysis for LiPSs. In addition, the relationship between defective structure and adsorptive–catalytic properties was reasonably interpreted through a series of electrochemical measurements and density functional theory (DFT) calculations. Oxygen vacancies could reduce the band gap of Bi_3_TaO_7_ with increased intrinsic conductivity and supply suspended unsaturated bonds around the vacancies to increase binding energy, At the same time, the multidimensional carbon structure prevents the agglomeration and volume expansion of active sulfur. The developed sulfur electrode with these superior features offers remarkable rate capability and cyclability with a low-capacity fading rate of 0.047% per cycle, and achieves a high areal capacity of 10.2 mAh cm^−2^ under practically relevant sulfur loading and electrolyte content with a high practical specific energy of near 300 W h kg^−1^. This study shed light on the mechanism for improving the performance through defect engineering and providing a new vision for advancing the practical application of Li-S batteries.

## Figures and Tables

**Figure 1 nanomaterials-12-03551-f001:**
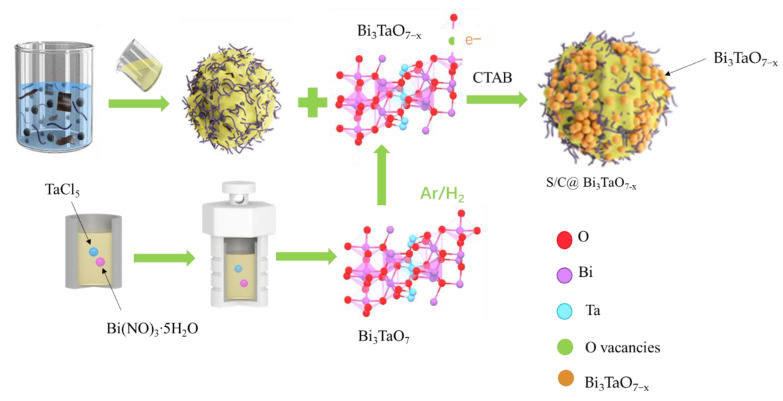
Schematic illustration of the synthesis of S/C@Bi_3_TaO_7−x_.

**Figure 2 nanomaterials-12-03551-f002:**
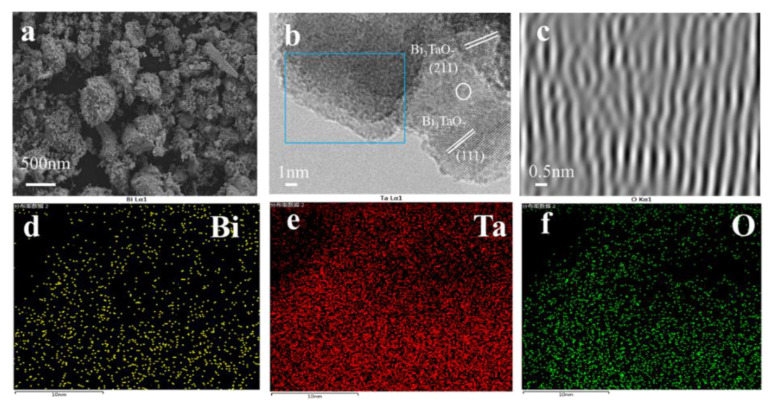
(**a**) SEM images. (**b**) TEM images. (**c**) FFT pattern image of Bi_3_TaO_7−x_ in the selected area. (**d**–**f**) Corresponding EDS elemental mappings of Bi_3_TaO_7−x_.

**Figure 3 nanomaterials-12-03551-f003:**
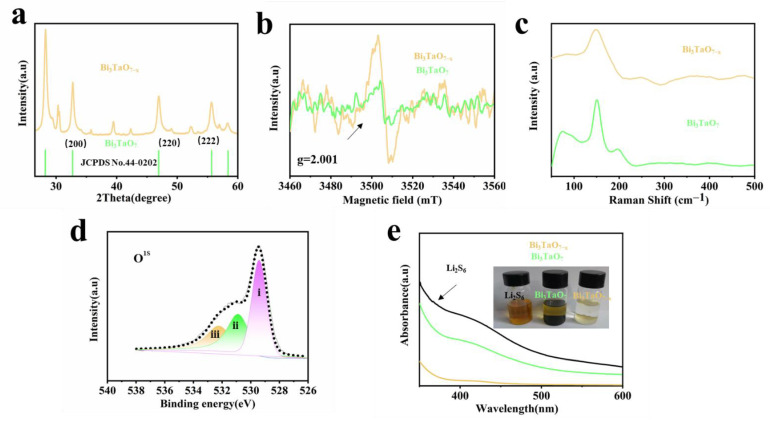
(**a**) XRD patterns of the Bi_3_TaO_7−x_ and Bi_3_TaO_7_. (**b**) EPR spectra of the Bi_3_TaO_7−x_ and Bi_3_TaO_7_. (**c**) Raman spectra of the Bi_3_TaO_7−x_ and Bi_3_TaO_7_. (**d**) High-resolution O1s XPS spectra of the Bi_3_TaO_7−x_. (**e**) UV–vis spectra of Li_2_S_6_ solution with Bi_3_TaO_7−x_ and Bi_3_TaO_7_ after treatment for 12 h. Insert image corresponds to the optical photograph of the above solution.

**Figure 4 nanomaterials-12-03551-f004:**
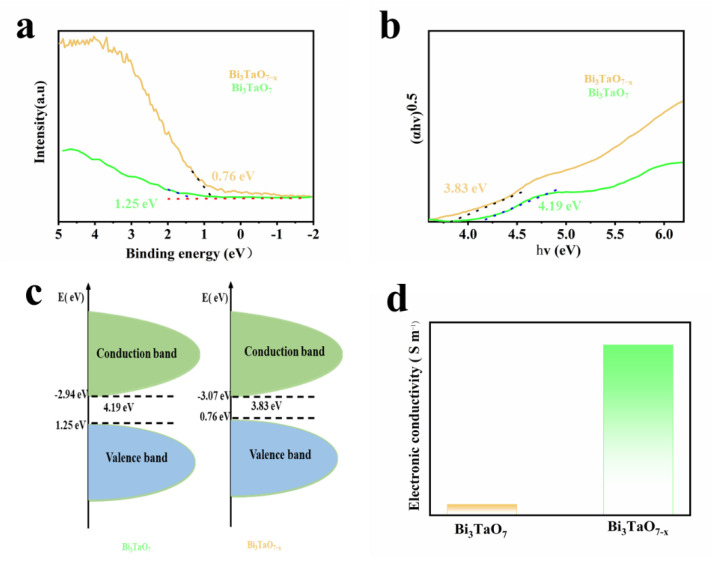
(**a**) Valence band XPS spectra, (**b**) Kubelka–Munk plot, and (**c**) band diagram of Bi_3_TaO_7−x_ and Bi_3_TaO_7_. (**d**) Electrical conductivity of Bi_3_TaO_7−x_ and Bi_3_TaO_7_.

**Figure 5 nanomaterials-12-03551-f005:**
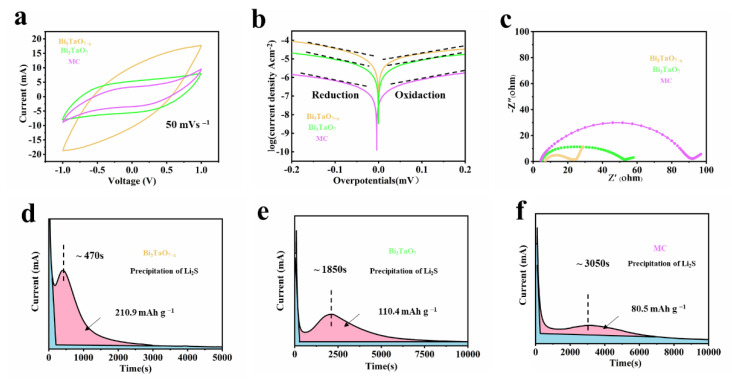
(**a**) Cyclic voltammetry curves for Li_2_S_6_ symmetric cells employing Bi_3_TaO_7−x_ and Bi_3_TaO_7_. (**b**) Tafel plots of Li_2_S_6_ symmetric cells. (**c**) EIS spectra of Li_2_S_6_ symmetric cells. (**d**–**f**) Precipitation profiles of Li_2_S.

**Figure 6 nanomaterials-12-03551-f006:**
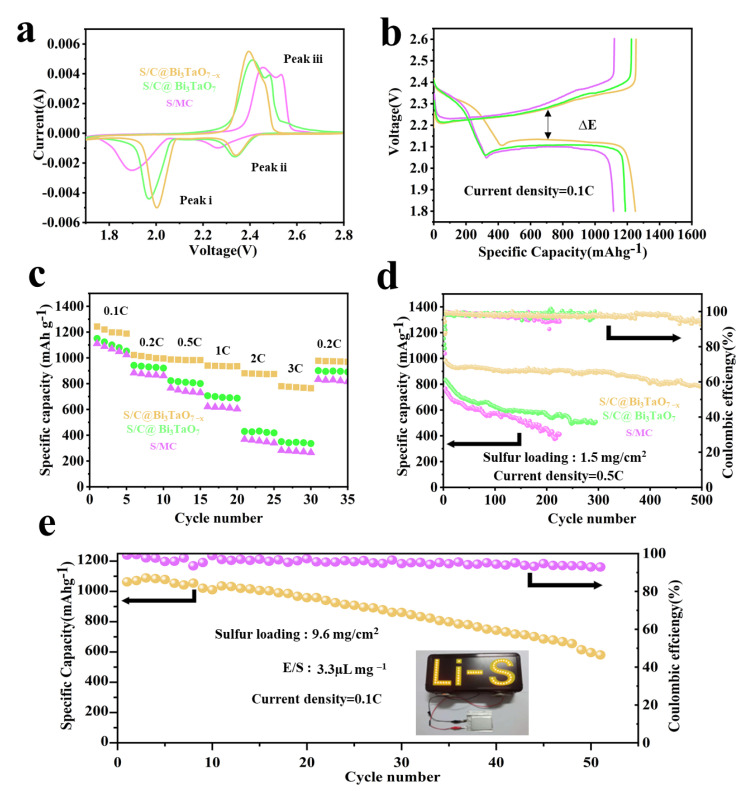
(**a**) CV profiles of coin cells at a scan rate of 0.1 mVs^−1^ employing Bi_3_TaO_7−x,_ Bi_3_TaO_7_, and MC electrodes. (**b**) Charge–discharge profiles at 0.1 C of various sulfur electrodes. (**c**) Rate performance of different sulfur electrodes. (**d**) Long cycling performances of Bi_3_TaO_7−x,_ Bi_3_TaO_7_, and MC electrodes at 0.5 C. (**e**) Cycle performance of the Bi_3_TaO_7−x_-based pouch cell at 0.1 C.

**Figure 7 nanomaterials-12-03551-f007:**
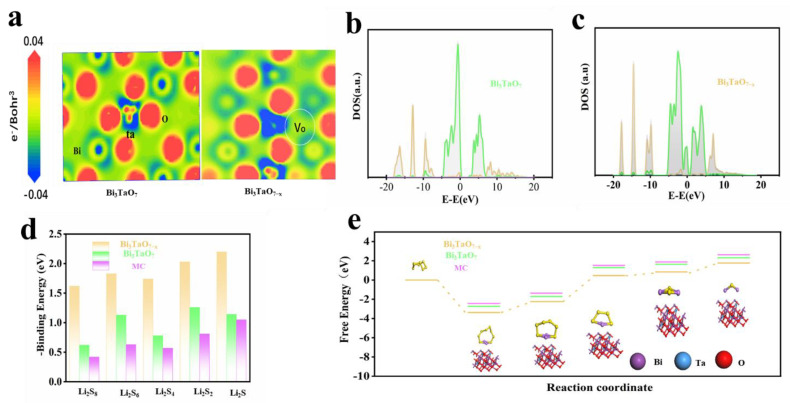
(**a**) The section of charge density difference of Bi_3_TaO_7_ and Bi_3_TaO_7−x_. (**b**,**c**) The calculated DOS for Bi_3_TaO_7_ and Bi_3_TaO_7−x_. (**d**) The binding energies between three samples and different LiPSs. (**e**) The Gibbs free energy from S_8_ to Li_2_S on surface Bi_3_TaO_7−x_, Bi_3_TaO_7_, and MC.

## Data Availability

Data presented in this article are available on request from the corresponding author.

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
