# Peer review of "Oxygen Vacancies in Bismuth Tantalum Oxide to Anchor Polysulfide and Accelerate the Sulfur Evolution Reaction in Lithium–Sulfur Batteries"

_nanomaterials, 2022, doi:10.3390/nano12203551_

Round 1
Reviewer 1 Report
Summary:
The Research Article nanomaterials 1947188 titled, “Oxygen vacancies in Bismuth tantalum oxide for polysulfides anchoring and accelerating the sulfur evolution reaction in lithium-sulfur batteries” reports the synthesis of oxygen vacancies Bi4TaO7-x nanoparticles and its application as a protection coating on a sulfur cathode. The sulfur cathode prepared with multidimensional carbon as Super P/CNT/Graphene/sulfur has an active material host and electron transfer matrix to couple with the outer-coated oxygen vacancies Bi4TaO7-x with improved electronic conductivity and strong affinity for polysulfides. The improved electrochemical reaction capability and stability allow the use of high sulfur loading cathode for an areal capacity of 10.20 mAh cm-2.
General comment:
This research shows the design of material and substrate of the lithium-sulfur battery cathode. A minor revision is therefore suggested to provide some necessary data and parameters. Hope the authors feel the comment useful. Thank you.
Comments:
(1) Different cells are used for the electrochemical analysis and performance. The electrochemical properties are different. Thus, the Experimental section is suggested to give the corresponding sulfur loading, sulfur content, and electrolyte/sulfur ratio of each cell. The sulfur content is suggested to be reported based on the whole cathode for a better understanding. The electrolyte/sulfur ratio is suggested to use the same unit as μL mg-1.
[Suggestion] Please revise Experimental section and give the necessary information correctly.
(2) In the material analysis, the Bi3TaO7-x’s minor peak should be identified. The minor peaks of Bi3TaO7-x do not all belong to Bi3TaO7. Why? The phase and purity of the Bi3TaO7-x are suggested to be discussed.
[Suggestion] Please analyze and discuss the Bi3TaO7-x material.
(3) The visualized adsorption experiment shows different information and result in the Experimental and Results sections. The MC sample and data are missing. It is also suggested to provide the as-prepared sample before resting.
[Suggestion] Please provide the missing MC data to the revised visualized adsorption experiment.
(4) The CV results show two anodic peaks in the Bi3TaO7 and MC electrodes. The electrode employing Bi3TaO7-x shows only one anodic peak. Why? The use of Bi3TaO7-x to trap polysulfides seems to cause the decrease of discharge/charge efficiency. Why? The size and preparation of pouch cell are suggested to be reported
[Suggestion] Please give some comment and suggestion toward the electrochemical data.
(5) Some recent reference using composite and carbon host in trapping polysulfides in the lithium-sulfur battery cathode are suggested to support the discussion in the introduction (Metallic: Batteries & Supercaps 2022, 5, e202100323; Oxides: ACS Sustainable Chemistry & Engineering 2022, 10, 9254; Transition‐Metal Oxides: ChemElectroChem 2022, 9, e202200374; Oxides: Materials Chemistry and Physics 2020, 255, 123484)
[Suggestion] Please support the discussion with reference citation and summarize the development trend and the potential applications.
Author Response
Point 1: Different cells are used for the electrochemical analysis and performance. The electrochemical properties are different. Thus, the Experimental section is suggested to give the corresponding sulfur loading, sulfur content, and electrolyte/sulfur ratio of each cell. The sulfur content is suggested to be reported based on the whole cathode for a better understanding. The electrolyte/sulfur ratio is suggested to use the same unit as μL mg-1.
The electrolyte density is 1.09, which is close to 1. I use the same unit of μL mg-1 now.
Point 2: In the material analysis, the Bi3TaO7-x’s minor peak should be identified. The minor peaks of Bi3TaO7-x do not all belong to Bi3TaO7. Why? The phase and purity of the Bi3TaO7-x are suggested to be discussed.
Due to the existence of oxygen vacancies, a very slight shift to a high angle can be seen.
Point 3: The visualized adsorption experiment shows different information and result in the Experimental and Results sections. The MC sample and data are missing. It is also suggested to provide the as-prepared sample before resting.
Based on our previous experimental experience, the weak polarity of pure carbon materials can only confine sulfur species by physical blocking instead of chemical adsorption, which means that the diffusion of polysulfides cannot be fully eliminated.
Point 4: The CV results show two anodic peaks in the Bi3TaO7 and MC electrodes. The electrode employing Bi3TaO7-x shows only one anodic peak. Why? The use of Bi3TaO7-x to trap polysulfides seems to cause the decrease of discharge/charge efficiency. Why? The size and preparation of pouch cell are suggested to be reported
The electrode employing Bi3TaO7-x also shows two anodic peak, different laps have certain differences.
In the later stages of the long-cycle test, the shuttle effect leads to rapid capacity decay and low Coulombic efficiency for Li-S batteries.
Li-S pouch cell included six 5.6 cm × 7.2 cm cathode slices with a sulfur loading of nearly 9.6 mg.cm-2 (two sides) and an E/S ratio of 3.3 µL mg-1
Point 5: Some recent reference using composite and carbon host in trapping polysulfides in the lithium-sulfur battery cathode are suggested to support the discussion in the introduction (Metallic: Batteries & Supercaps 2022, 5, e202100323; Oxides: ACS Sustainable Chemistry & Engineering 2022, 10, 9254; Transition‐Metal Oxides: ChemElectroChem 2022, 9, e202200374; Oxides: Materials Chemistry and Physics 2020, 255, 123484)
Reviewer 2 Report
The authors reported that Bi3TaO7-x with oxygen vacancies was synthesized, which affords the synergistic functions of physical confinement, chemical anchoring, and excellent electrocatalysis for LiPSs in lithium sulfur batteries. The relationship between defective structure and superior electrochemical performance was investigated by experiment and DFT. Remarkably, the coin cells and pouch cells show promising performance. Therefore, I would like to recommend to publish. The additional issues are as following:
1. The XRD pattern of Bi3TaO7 sample is missing In Fig. 3a. In addition, there are some unmatched peaks in the XRD pattern of Bi3TaO7-x, where do these peeks come from?
2. Based on the charge compensation, the valence of Bi or Ta would change if there are oxygen vacancies in Bi3TaO7-x. Please provide the explanation.
3. The related literature (J. Power Source 2022, 520, 230885; Energy Technol. 2021, 9, 2001057; Energy Technol. 2019, 7, 1900583.) was suggested to cite.
4. Please check the y-axis unit of Fig. 6a.
5. The company of chemicals should be provided. The scale of Fig. 2c should be provided.
Author Response
The authors reported that Bi3TaO7-x with oxygen vacancies was synthesized, which affords the synergistic functions of physical confinement, chemical anchoring, and excellent electrocatalysis for LiPSs in lithium sulfur batteries. The relationship between defective structure and superior electrochemical performance was investigated by experiment and DFT. Remarkably, the coin cells and pouch cells show promising performance. Therefore, I would like to recommend to publish. The additional issues are as following:
Point 1: The XRD pattern of Bi3TaO7 sample is missing In Fig. 3a. In addition, there are some unmatched peaks in the XRD pattern of Bi3TaO7-x, where do these peeks come from?
Due to the existence of oxygen vacancies, a very slight shift to a high angle can be seen.
Point 2: Based on the charge compensation, the valence of Bi or Ta would change if there are oxygen vacancies in Bi3TaO7-x. Please provide the explanation.
The shift in the peaks of Ta 4f demonstrates change in the surrounding coordination environment.
Point 3: The related literature (J. Power Source 2022, 520, 230885; Energy Technol. 2021, 9, 2001057; Energy Technol. 2019, 7, 1900583.) was suggested to cite.
I have cited the related literature.
Point 4: Please check the y-axis unit of Fig. 6a.
I have checked the the y-axis unit of Fig. 6a.
Point 5: The company of chemicals should be provided. The scale of Fig. 2c should be provided.
I have provided the company of the chemicals and the scale of Fig. 2c.
Reviewer 3 Report
The article entitled “Oxygen vacancies in Bismuth tantalum oxide for polysulfides 2 anchoring and accelerating the sulfur evolution reaction in lithium-sulfur batteries” is interesting and advanced research work in the battery research.
The article is well written with good experimental results. The improved electrical conductivity via Oxygen vacancies is interesting. The article may be considered for the publication with minor revision.
However, I would like to ask few clarifications and suggestions
1. The line from 63 to 65 is confusing. Authors may clearly write the sentences “Herein, Bi3TaO7 with oxygen vacancies is used as an example to establish the relationship between oxygen vacancy and adsorptive-catalytic properties. Based on the above analysis,”
2. In the preparation of cathode, the fraction or ratio of P, CNT, and Graphene is not explained. Author may provide the more details about the cathode material.
3. It will be easy understanding if the authors named all the materials (MC or carbon materials) in the schematic illustration.
4. Authors provided EDS mapping (Fig.2d-f). However, it is necessary to add the base image with same scale bar or the merged image with all the elements to locate the positions of the element in the sample. Like S3 image in the supplementary.
5. Does E/S ratio need a unit? It may also be the numbers alone.
6. The sample codes in figure 5 is not clear. Author may improve that.
7. The cathodic peaks at 2.03 and 2.35 V intensity is varying for different materials. Author may provide detailed discussion on the LiPS conversion in relation with the cathodic and anodic peak.
Author Response
Point 1: The line from 63 to 65 is confusing. Authors may clearly write the sentences “Herein, Bi3TaO7 with oxygen vacancies is used as an example to establish the relationship between oxygen vacancy and adsorptive-catalytic properties. Based on the above analysis,”
I have revised these sentences.
Point 2: In the preparation of cathode, the fraction or ratio of P, CNT, and Graphene is not explained. Author may provide the more details about the cathode material.
A certain amount of Super P, CNT, Graphene (the weight ratio of Super P: CNT: G: =2:2:1) was ball-milled to get the uniform slurry.
Point 3: It will be easy understanding if the authors named all the materials (MC or carbon materials) in the schematic illustration.
I have accepted your suggestions
Point 4: Authors provided EDS mapping (Fig.2d-f). However, it is necessary to add the base image with same scale bar or the merged image with all the elements to locate the positions of the element in the sample. Like S3 image in the supplementary.
I have located the position of the element in the sample.
Point 5: Does E/S ratio need a unit? It may also be the numbers alone.
The electrolyte density is 1.09, which is close to 1. I use the same unit of μL mg-1 now.
Point 6: The sample codes in figure 5 is not clear. Author may improve that.
I have improved that.
Point 7: The cathodic peaks at 2.03 and 2.35 V intensity is varying for different materials. Author may provide detailed discussion on the LiPS conversion in relation with the cathodic and anodic peak.
It is found that the S/C@Bi3TaO7-x cathode displays two typical cathodic peaks at 2.03V(peak ⅰ) and 2.35V(peak ⅱ), which are corresponding to the reduction of sulfur into high-order soluble LiPSs and the subsequent conversion of the LiPSs insoluble Li2S2/ Li2S. The anodic peaks at 2.38V(peak ⅲ ) are associated with the reverse oxidation conversion from Li2S to LiPSs and finally to sulfur
